# Identification of Intermetallic Phases Limiting the Growth of Austenite Grains in the Low-Pressure Carburizing Process

Konrad Dybowski [ID] and Leszek Klimek *[ID]

Institute of Materials Science and Engineering, Lodz University of Technology, 90-537 Lodz, Poland;
konrad.dybowski@p.lodz.pl
* Correspondence: leszek.klimek@p.lodz.pl

**Abstract:** This article presents the results of a study to identify intermetallic phases whose role is to limit austenite grain growth in the low-pressure carburizing process. A drawback of high-temperature low-pressure carburizing is the austenite grain growth during the process. Using low-pressure carburizing with pre-nitriding technology (PreNitLPC®) offers the possibility of reducing austenite grain growth. This technology involves the application of doses of ammonia during the heating stage of the steel, at the carburizing temperature, to introduce nitrogen into the surface layer of the steel and to form nitrides. It is these phases that cause restrictions on austenite grain growth during carburizing. The research carried out in this article was aimed at identifying these phases. The research was carried out on one of the basic steels used for carburizing—16MnCr5 steel. The carburizing of this steel with and without pre-nitriding was performed, followed by an evaluation of the austenite grain size after these processes and the identification of the intermetallic phases present in the surface layer of the steel.

**Keywords:** thermo-chemical treatment; low-pressure carburizing; pre-nitriding; intermetallic phases

## 1. Introduction

Carburizing is a well-known thermo-chemical treatment technology and continues to be the leading surface treatment for steel. It is estimated that about 60–70% of the mass share of steel parts produced for the automotive industry are treated with this technology. This is due to the fact that carbon is the primary alloying additive added to iron, having the strongest effect on increasing its mechanical properties. Low-pressure carburizing (also called vacuum carburizing) is the most modern variation of gas carburizing technology. It successfully competes with other carburizing methods due to its many benefits. These include the reduction in emissions of harmful substances into the atmosphere, the clean surface of the charge after the process, the reduction in hardening deformations and the possibility of increasing the process temperature, which translates into an economic effect in the form of a significant reduction in the duration of the entire process [1–3]. This is due to the process being carried out in an atmosphere with a much higher carbon potential and the possibility of using higher carburizing temperatures than traditionally used. Currently, the temperatures used in low-pressure carburizing can reach up to 1050 °C, which significantly shortens the process and makes this technology unrivaled compared to conventional methods [1,4,5]. The most important limitation of using such high temperatures is the significant growth of austenite grains in the carburized steel [6]. This results in the formation of coarse-grained martensite during hardening, leading to a reduction in the surface layer's technological properties. This can be prevented by using steels with alloying elements that limit grain growth [6–16]. Research presented in the literature indicates that alloy additions and modifications of the carburizing process have the effect of controlling the size of austenite grains during this process at various temperatures [10,17]. It has been shown that small particles of nitrides and carbides can limit grain growth [18]. In steel with the

micro-addition of titanium, intermetallic phases TiNb, TiN, and (Ti,Nb)C are precipitated, which play a key role in inhibiting the growth of austenite grains during the carburizing process [18,19]. However, the final effect of grain refinement is also influenced by the size of these precipitates. If these precipitates are larger than 100 nm, the effect of grain refinement weakens [20,21]. Many of the works analyzed indicate the control of austenite grain size using TiN particles, because titanium nitrides have excellent thermal stability [10,22]. Other studies show that AlN precipitates play a similar role to titanium-based precipitates. It has been shown that the Al and N content have a beneficial effect on obtaining a fine-grained structure [23–25]. Generally, nitrides are more temperature-stable than carbides, but the formation of nitrides and carbonitrides in steel is limited by the low content of free N [26–29]. To sum up, the use of steel with micro-additives reduces the growth of austenite grains, but increases material costs. In addition, most commercially available steels for carburizing do not contain these elements (they are added on demand) or, like nitrogen, they are introduced into these steels in an uncontrolled manner during the metallurgical process. There is also the danger of large inclusions based on these additives, which can reduce fatigue life [30,31].

Another way is to use appropriate thermal treatments after carburizing [32,33], but this, in addition to generating additional costs, results in a longer total processing time and, consequently, little justification for using high carburizing temperatures. An alternative solution may be the use of low-pressure carburizing with pre-nitriding (a technology known as PreNitLPC®) [2,3,34–36]. This technology, compared to conventional low-pressure carburizing, differs in the nitrogen additionally introduced into the steel. During heating to carburizing temperatures in the temperature range of 400 to 700 °C, ammonia is dosed into the vacuum furnace chamber. Ammonia dosing is carried out continuously over the temperature range given above. The amount of nitrogen supplied can be controlled by the amount of ammonia flow and the possible change in the heating rate. The temperature range at which ammonia is dosed is not random. It is optimal from the point of view of both the conditions of catalytic decomposition of ammonia on the surface of the charge and the total duration of the carburizing process. The range of temperatures at which ammonia is dosed is optimal both from the point of view of optimal conditions for the catalytic decomposition of ammonia, and for the total duration of the carburizing process. Dosing ammonia at the stage of heating the steel to the carburizing temperature does not increase the total process time; on the contrary, due to the possibility of using higher temperatures, it is possible to significantly reduce the process time [2,3]. The given ammonia dosing temperature range is also advantageous from the point of view of the possibility of the uncontrolled activation of the carbon deposit present in the chamber of the carburizing furnace. This constitutes an additional source of carbon and makes it impossible to precisely control the carburizing [36]. Ammonia dissociates catalytically on the steel surface and is a source of nitrogen atoms, which then saturate the surface layer of the steel. The nitrogen present in the steel causes the formation of nitrides, which affects the subsequent reduction of austenite grain growth. The results presented in this article confirm the role of nitrogen in the mechanism of limiting the growth of austenite grains in low-pressure carburizing PreNitLPC® technology. They indicate that nitrogen-forming intermetallic phases—nitride phases with one of the elements found in steel—make this possible. It has been shown that increasing the temperature by 80 °C did not affect the coarseness of the austenite, and in fact the grain size obtained at a higher temperature was smaller. It has also been proven that increasing the temperature does not negatively affect the structure of the surface layer and mechanical properties (hardness) obtained as a result of carburizing.

## 2. Materials and Methods

Steel cylindrical samples with dimensions of 30 mm × 25 mm, made of 16MnCr5 steel in a normalized condition, were used for the tests. The chemical composition of the steel is given in Table 1. After cutting, the flat surfaces were ground on a bench grinder equipped with a grinding wheel made of cubic boron nitride (CBN). The first batch of samples was

subjected to low-pressure carburizing (LPC) at a temperature of 920 °C. The second batch of specimens from the same steel was subjected to low-pressure carburizing with pre-nitriding (PreNitLPC®) at 1000 °C. The adopted criteria for the carburized layer are given in Table 2. Multi-stage carburizing was used, consisting of alternating boost and diffusion stages. The time for each process stage to obtain the assumed carbon concentration profile was selected based on simulations of the carburizing process performed with SimVaCPlus v.3.0 software. This program calculates the optimal time of the carburizing process for a given temperature and carburizing layer criteria (required carbon surface concentration, effective layer thickness). The total process time is divided into successive stages of carbon saturation and diffusion. Therefore, as a result of the operation of this program for the given parameters of the carburized layer and temperature (Table 2), an optimal division of the saturation/diffusion duration was obtained.

**Table 1.** Chemical composition of 16MnCr5 steel used for the study [wt%].

| C | Mn | Si | P | S | Ni | Cr | Cu | Mo | Al | Fe |
|---|----|----|---|---|----|----|----|----|----|----|
| 0.16 | 1.21 | 0.37 | 0.022 | 0.030 | 0.19 | 0.96 | 0.16 | 0.03 | 0.027 | rest |

**Table 2.** Process parameters for the adopted carburized layer criterion.

| Carburizing Temperature | Temperature Range for Ammonia Feeding | Surface Concentration of Carbon | Effective Layer Thickness (ECD) | Total Layer Thickness (TCD) | Process Time Boost/Diffusion |
|---|---|---|---|---|---|
| [°C] | [°C] | [wt%] | [mm] | [mm] | [min.] |
| 920 | - | 0.75 | 0.6 | 1.0 | 5/10, 4/16, 4/23, 3.5/30, 3/42, 3/40 |
| 1000 | 400–700 | | | | 4.5/6, 3/16, 2/17 |

The carburizing atmosphere was obtained from the thermal dissociation of a hydrocarbon mixture—ethylene and acetylene, diluted with hydrogen at a 2:2:1 ratio, fed into the furnace chamber at variable pressure from 300 to 800 Pa—while nitrogen in the pre-nitriding option was obtained from the catalytic decomposition of ammonia, feeding ammonia into the furnace chamber during the heating of the charge to reach a carburizing temperature at a temperature range from 400 °C to 700 °C and pressure of 2.6 kPa. The heating rate in both carburizing cases was the same and amounted to 10 °C/min. After carburizing, in both cases, hardening was applied from 860 °C in nitrogen at 1.2 MPa and low-temperature tempering at 180 °C.

Metallographic tests of the carburized steels were then carried out. The microstructure of the steels was studied and the size of austenite grains formed at 920 °C during low-pressure carburizing (LPC) and at a temperature of 1000 °C during low-pressure carburizing with preliminary nitriding (PreNitLPC®) was determined. These studies were performed using an Eclipse MA 2000 optical microscope from Nikon (Tokyo, Japan). The microstructure of the top steel layer was examined on transverse metallographic sections. The preparation of samples for testing consisted of cutting the sample using a metallographic cutter, embedding it in thermosetting resin, and grinding the cut surface using AlOx sandpaper with grit from 200 to 1200. After grinding, polishing was carried out in a water suspension of aluminum oxide. Then, after washing in deionized water and drying, they were etched in the Mi1Fe reagent. The grain size of austenite was determined in accordance with the ISO 643 standard. The grain size measurement in the surface layer of 16MnCr5 steel was carried out on cross-sections. The metallographic grind was etched with an aqueous solution of picric acid with a surfactant and heated to 70 °C to intensify the etching process.

A comparison study of the carbon concentration gradient in the surface layer of carburized steel was also performed using Leco's GDS 850A spectrometer (LECO Corporation, St.

Joseph, MI, USA). To evaluate the correctness of the carburizing processes carried out, microhardness distributions of the surface layer from the surface toward the core of 16MnCr5 steel after the LPC and PreNitLPC® processes were also performed and compared with each other. Microhardness measurements were made on cross-sections of the carburized steel, using the Vickers method with a CLEMEX JS-2000 microhardness tester (Clemex Tech USA Corp, Pkwy, Fenton, MI, USA) under a load of 0.98 N.

The next stage of the study was to determine the mechanisms of the observed phenomenon causing a reduction in the growth of austenite grains in the surface layer of steel after low-pressure carburizing with pre-nitriding. For this purpose, the nitrogen concentration gradient in the surface layer of carburized steel was determined. The study was carried out using the conductivity method according to ISO 10720, using a TCH600 nitrogen analyzer from Leco (LECO Corporation, St. Joseph, MI, USA). Subsequently, studies were carried out to identify nitrogen-based intermetallic phases, which were the reason for the limiting of the growth of austenite grains in the carburizing process with PreNitLPC® technology. The research was carried out using FEI's ultra-high-resolution scanning transmission electron microscope (S/TEM) TITAN 80–300 (FEI Company, Hillsboro, OR, USA), applying electron accelerating voltages in the range of 100–200 kV. Thin films for testing were prepared using an ion (Ga+) cannon mounted on FEI's Quanta 3D 200i scanning electron microscope (FEI Company, Hillsboro, OR, USA). Chemical composition studies were performed using a microscope-mounted EDS X-ray analyzer from EDAX and an EELS spectrometer (FEI Company, Hillsboro, OR, USA) in the form of a GIF Tridiem 863P electron kinetic energy filter from Gatan.

## 3. Results

### 3.1. Carbon Concentration Gradient and Microhardness Distribution in the Surface Layer of Precipitates of 16MnCr5 Steel after LCP and PreNitLPC® Processes

The graph depicted in Figure 1 compares two carbon concentration profiles in the surface layer of 16MnCr5 steel obtained by carburizing with LPC and PreNitLPC® technologies. As can be seen, in both cases, i.e., with both LPC carburizing carried out at 920 °C and PreNitLPC® carburizing at 1000 °C, almost identical carbon concentration distributions were obtained. As assumed in Table 2, a surface concentration of 0.75 wt% C, an effective layer thickness of ECD = 0.6 mm, and a total layer thickness of TCD = 1.0 mm were obtained. Obtaining the same parameters of the carburized layer in both carburizing processes provides an opportunity to further compare these processes in terms of austenite grain size.

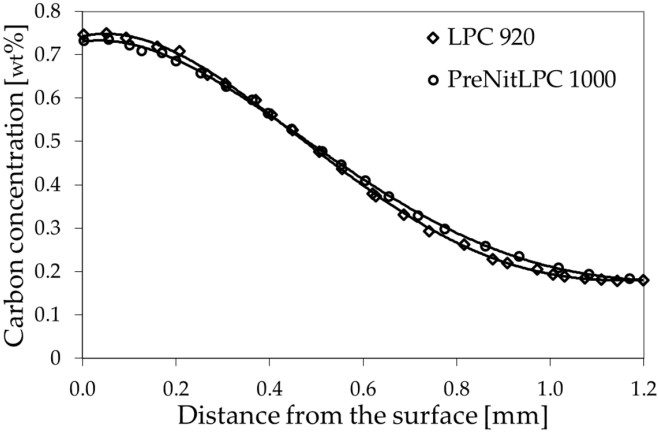

**Figure 1.** Carbon concentration gradient in the surface layer of 16MnCr5 steel after LCP and PreNitLPC® processes.

Another comparison of the efficiency of the low-pressure carburizing process and the PreNitLPC® process concerned the microhardness distribution of the surface layer of

16MnCr5 steel (Figure 2). Here, too, almost identical hardness distributions were obtained in both cases. Slight differences were found in the near-surface region. The surface hardness of 16MnCr5 steel after the LPC process was slightly lower than that of the same steel after the PreNitLPC® process. The first case yielded about 700 HV01, and the second about 750 HV01. Further, at a depth of about 0.2 mm from the surface, these hardnesses equalized and further, to the core, the hardness distribution was identical. The slightly lower hardness for the LPC process may be due to the larger austenite grain (as shown later in the article). A larger austenite grain, after hardening, results in coarse-grained martensite and more retained austenite, which affects the hardness.

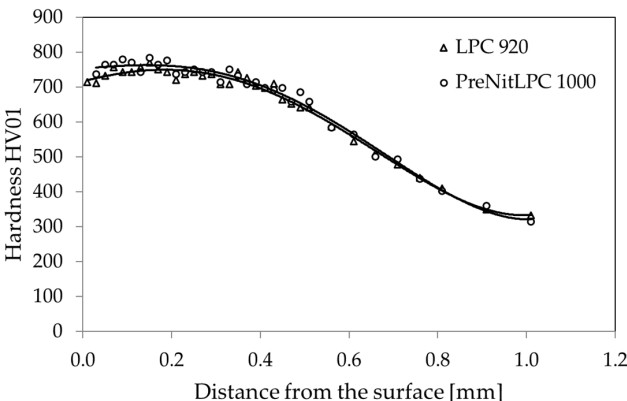

**Figure 2.** Hardness distribution in the surface layer of 16MnCr5 steel after the LPC process at 920 °C and PreNitLPC® at 1000 °C.

### 3.2. Nitrogen Concentration Gradient in the Surface Layer of 16MnCr5 Steel after the PreNitLPC® Process

For samples of 16MnCr5 steel subjected to low-pressure carburizing with pre-nitriding (PreNitLPC®), nitrogen distribution from the surface to the core of the steel was measured (Figure 3). As can be seen, the surface concentration of this element at the surface is about 0.03 wt% and decreases gradiently to the core, reaching a value of about 0.012 wt% N in the core. The total thickness of the diffusion layer is about 1.0 mm. This means that it is comparable to the total thickness of the carburized layer.

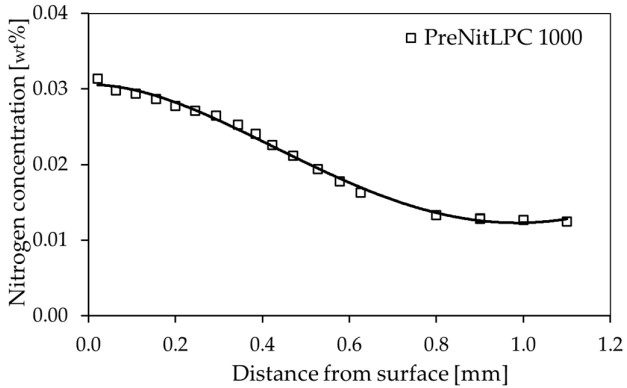

**Figure 3.** Nitrogen concentration gradient in the surface layer of 16MnCr5 steel after the PreNitLPC® process.

### 3.3. Microstructure Studies of the Surface Layer of 16MnCr5 Steel after LPC and PreNitLPC® Processes—Determination of Austenite Grain Size

After the low-pressure carburizing process of LPC and PreNitLPC®, metallographic examinations of the surface layer of processed steels were performed (Figure 4). The microstructure of these steels consists of martensite and a small amount of retained austenite.

There was a correct structural structure, typical of steel subjected to carburizing, hardening and low-temperature tempering. The difference in structure between low-pressure carburized steel (Figure 4a) and low-pressure carburized steel with preliminary nitriding (Figure 4b) is only related to the length of the martensite needles. This, in turn, results from the grain size of the original austenite. The austenite grain diameter after the PreNitLPC® process is smaller, as shown below.

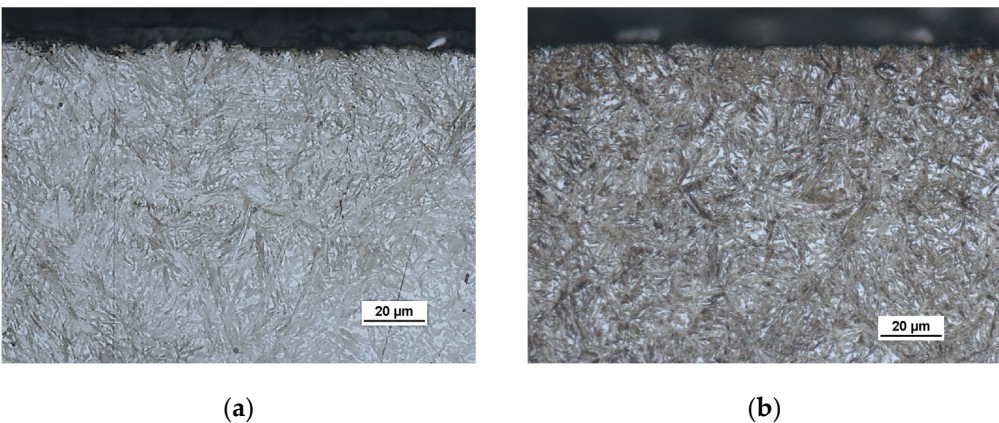

(**a**)        (**b**)

**Figure 4.** Microstructure of the surface layer of 16MnCr5 steel after low-pressure carburizing at 920 °C (**a**) and after low-pressure carburizing at 1000 °C with pre-nitriding (**b**). Visible martensite with retained austenite.

The effect of pre-nitriding applied before carburizing can be seen in Figure 5. The microstructure images there show the difference in austenite grain size in the surface layer of 16MnCr5 steel that was formed by the LPC process (Figure 5a) and PreNitLPC® (Figure 5b). In both cases, the average austenite grain size was measured. The average grain diameter in the surface layer of 16MnCr5 steel after low-pressure carburizing was 14 μm, while after carburizing with pre-nitriding it was 10 μm. So even though the PreNitLPC® process was carried out at a temperature 80 °C higher than LPC, the austenite grain was about 30% smaller. Such an effect makes it possible to use a higher process temperature and thus reduce the carburizing time by a factor of almost four without the unfavorable growth of austenite grains and obtaining a coarse-grained martensite structure after quenching.

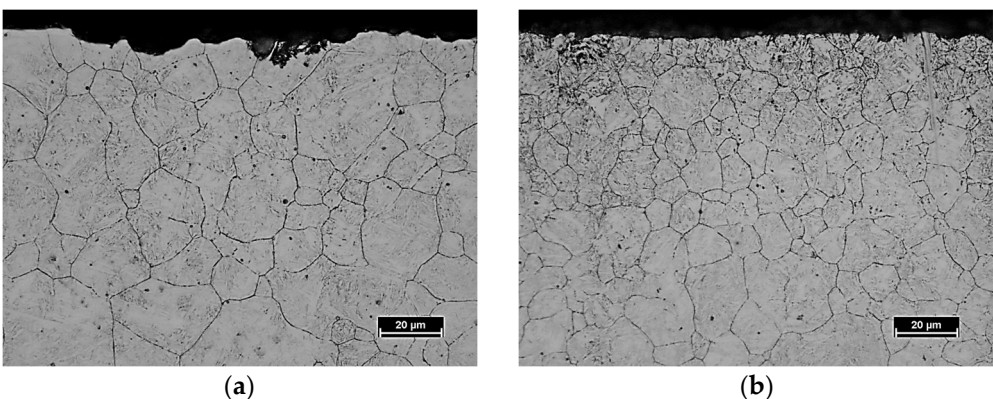

(**a**)        (**b**)

**Figure 5.** Microstructure of the surface layer of 16MnCr5 steel after low-pressure carburizing at 920 °C (**a**) and after low-pressure carburizing at 1000 °C with pre-nitriding (**b**). Visible austenite grain boundaries.

### 3.4. Identification of Intermetallic Phases in the Surface Layer of 16MnCr5 Steel after the PreNitLPC® Process

As a result of the examination of the structure of the surface layer of 16MnCr5 steel using a S/TEM microscope (FEI Company, Hillsboro, OR, USA), the presence of precipitates with a size of several dozen nanometers was found (Figure 6). A much larger amount of them was found in steel subjected to the low-pressure carburizing process with preliminary nitriding—PreNitLPC® (Figure 6b). In the surface layer of the same steel after the low-pressure carburizing process, LPC (Figure 6a), they were also present, but they were single particles.

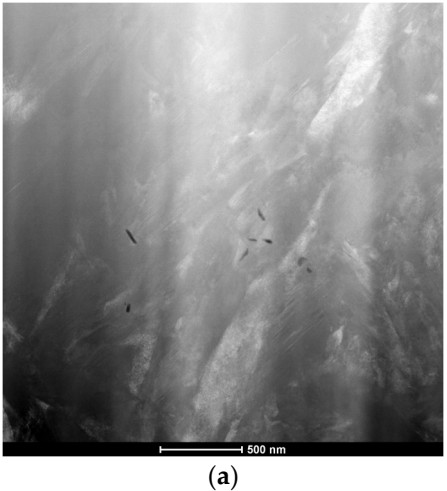 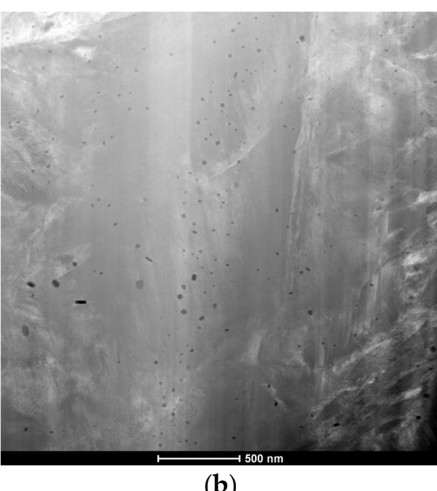

(**a**)           (**b**)

**Figure 6.** S/TEM images of inclusions observed in the structure of the surface layer of 16MnCr5 steel after low-pressure carburizing at 920 °C (**a**) and after low-pressure carburizing at 1000 °C with pre-nitriding (**b**).

A detailed quantitative analysis of these precipitates was performed using the MetIlo v. 9.07 program. The analysis was performed on ten S/TEM microscope images, covering an area of 3 × 3 µm. The methodology of quantitative measurements using the MetIlo program is described in the literature [37]. Before analysis, each image was subjected to image histogram normalization. As a result of this transformation, no data were lost—the photo had all the structural elements obtained in the image acquisition process. Using this transformation, we extended the image histogram to the entire tonal range, obtaining the effect of optimal contrast. The next step was the manual binarization of the gray image. As a result of these transformations, the optimal image contrast was obtained, thanks to which all structure elements were revealed in the photo. Image processing carried out in this way allowed for the precise determination of the number of examined inclusions in the photographs. The measurement results are presented in Table 3. They indicate that the number of inclusions observed in the surface layer of 16MnCr5 steel after the PreNitLPC® process was approximately 6–7 times higher than in this steel after carburizing with the LPC technology.

**Table 3.** The number of precipitations in the surface layer of steel after the low-pressure carburizing process (LPC) and after the low-pressure carburizing process with preliminary nitriding (PreNitLPC®).

|  | LPC | PreNitLPC® |
|---|---|---|
| Average number of precipitates | 8 | 54 |
| Standard deviation | 1.7 | 8.9 |

The results of tests to identify these precipitates are shown in Figures 7 and 8. As these show, the visible nanostructured precipitates in 16MnCr5 steel both after the PreNitLPC® (Figure 7a) process and after the LPC (Figure 8a) process are aluminum nitrides, AlN. Alloying additives such as manganese and silicon are dissolved in these intermetallic phases, as indicated by the elemental distribution maps (Figures 7b and 8b). So, the identified nano-precipitates are solutions of Mn and Si in aluminum nitride. Although many precipitates were identified in the steels studied, as the results presented here indicate, each time they were the same intermetallic phases formed based on aluminum and nitrogen. The article presents only two examples of identified precipitates—one in steel after the PreNitLPC® process and one after the LPC process.

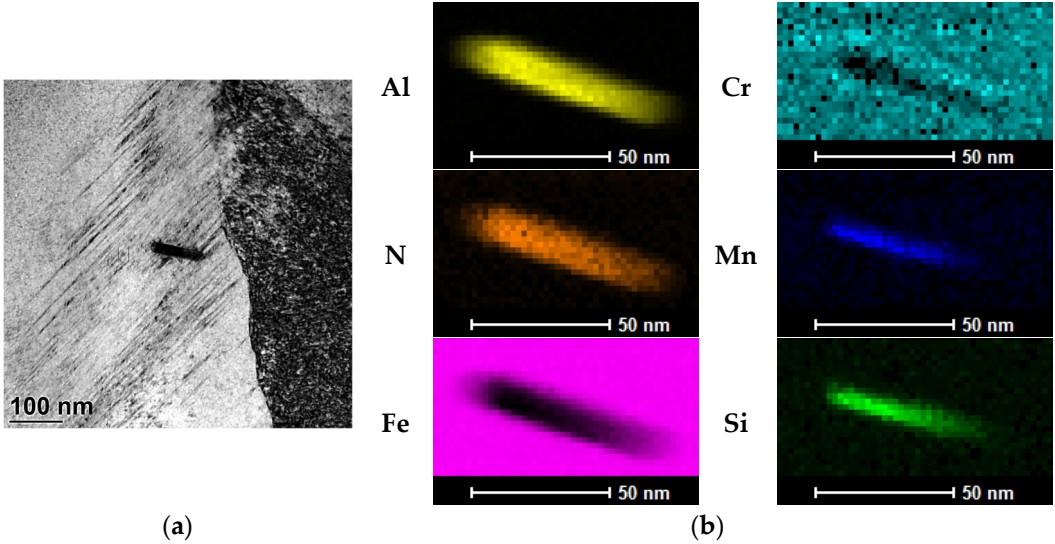

(**a**)           (**b**)

**Figure 7.** TEM image of the inclusion identified in the structure of the surface layer of 16MnCr5 steel after the PreNitLPC® process (**a**), and the elemental distribution map (**b**).

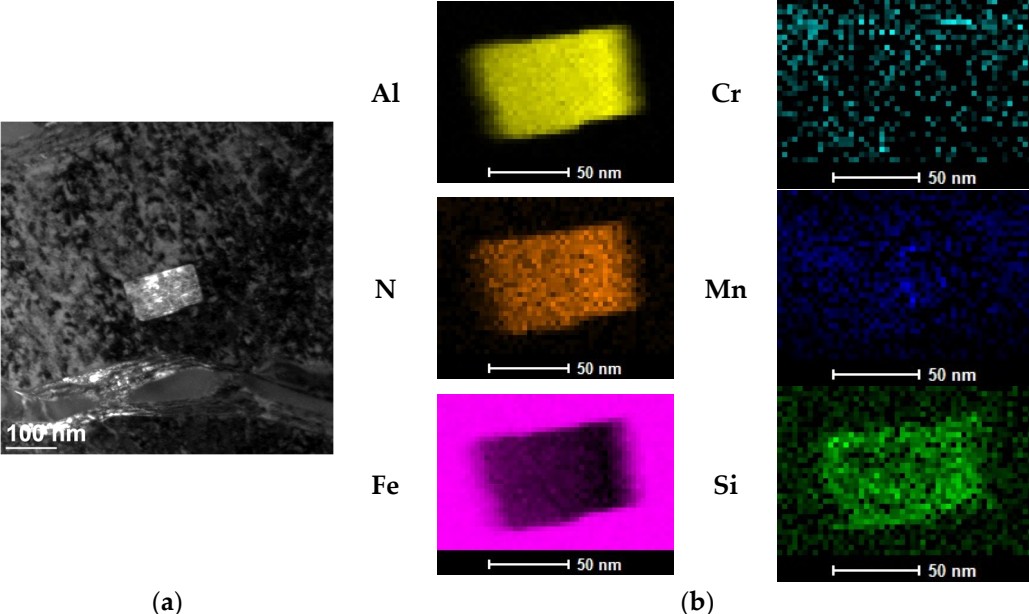

(**a**)           (**b**)

**Figure 8.** TEM image of the inclusion identified in the structure of the surface layer of 16MnCr5 steel after the LPC process (**a**), and the elemental distribution map (**b**).

## 4. Discussion and Conclusions

The method known from the literature to reduce the growth of austenite grains in the heat treatment processes of steels is through the use of intermetallic phases forming micro-additives in these steels. The presence of these phases effectively reduces the austenite grain size. A similar effect, without the need to use special steels with micro-additives, can be obtained, as demonstrated in this article, on ordinary carburizing steels. In this case, intermetallic phases are formed based on nitrogen introduced into the steel before it is saturated with carbon. Significantly, nitrogen in this case forms nitrides only with aluminum, even though other alloying additives with much higher content and equally high affinity for nitrogen are present. The relatively small amount of nitrogen introduced, as well as the low aluminum content, is sufficient to create nanometric intermetallic phases in an amount that exerts a limiting effect on austenite grain growth. It is also worth noting that the aluminum in this steel is not an alloying additive, but only an admixture, added at the deoxidation stage of this steel. Therefore, its presence is small, and the amount is uncontrolled because it is the residue not used to bind oxygen. In summary, the effect of low-pressure carburizing technology with preliminary nitriding (PreNitLPC®) allows the carburizing temperature to be increased by several tens of degrees Celsius, without the growth of austenite grains, and consequently for a fine-needle martensite structure to be obtained. This translates into significant reductions in the carburizing process time as a result of an increase in the carbon diffusion coefficient, which depends exponentially on temperature. When comparing the carburizing processes presented in this article, the reduction in carburizing time for the parameters assumed here was 127 min in favor of PreNitLPC®. The reduction in austenite grain size distribution consequently makes it possible, at higher carburizing temperatures, to obtain high-quality carburized layers with very good mechanical properties, comparable to those carburized at conventional carburizing temperatures.

**Author Contributions:** Conceptualization, K.D. and L.K.; methodology, K.D. and L.K.; validation, K.D.; formal analysis, K.D.; investigation, K.D.; resources, L.K. and K.D.; data curation, K.D. and L.K.; writing—original draft preparation, K.D.; writing—review and editing, L.K. and K.D.; visualization, K.D. and L.K.; supervision, K.D.; project administration, L.K. All authors have read and agreed to the published version of the manuscript.

**Funding:** This research received no external funding.

**Data Availability Statement:** The data presented in this study are available on request from the corresponding author. The data are not publicly available due to lack of access to storage space.

**Conflicts of Interest:** The authors declare no conflict of interest.

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
