# Peer review of "Identification of Intermetallic Phases Limiting the Growth of Austenite Grains in the Low-Pressure Carburizing Process"

_crystals, doi:10.3390/cryst13121683_

Round 1

Reviewer 1 Report

Comments and Suggestions for Authors

The technology of carburizing austenite at low pressure with pre-nitriding (PreNitLPC® technology) was proposed more than 10 years ago and showed the possibility of reducing grain growth. The reason for the restriction on the growth of austenite grains during carburization is the formation of  nanostructured precipitates  (intermetallic phases). The purpose of this article is to identify the mechanisms of formation of intermetallic phases. The article shows that the nanostructured precipitates are solutions of Mn, Si, and Cr in aluminum nitride. This can be considered a significant result of this article.

Question: «This translates into significant reductions in carburizing process time» (222-223). Can you specify the time?

Author Response

Question: «This translates into significant reductions in carburizing process time» (222-223). Can you specify the time?

  • when comparing the carburization processes presented in this article, the reduction in carburization time for the parameters assumed here was 127 minutes in favor of PreNitLPC®. The sentence was included in the article. (Line 293-295).

Reviewer 2 Report

Comments and Suggestions for Authors

This manuscript reports the microstructure difference of low-pressure carburizing process with and without pre-nitriding. It is clearly written, and I suggest acceptance with revisions.

-In abstract, line 14:

“on its basis”;  an incorrect word?  What do you intend to say? Base?

-Line 60: “intermetallic phases on its base”

Please clarify. Does the intermetallic phases form on the surface or inside core of the steel? What do you mean by base?

-Line 93: Delete “Assumed”.

-Figure 4:  it will be better if you can add two more images into this figure: the SEM grain images of the surface of the steel of samples with and without pre-nitriding. 

-Can the author provide SEM images to show where are the intermetallic precipitates located?  This image could be the same image in Figure 4.

- Figure 5 and Figure 6, and Line 196-197: “Therefore, it was decided to include in the article results two examples of the identified precipitates to show that they are of the same type.” 

                Please rephrase this sentence. It’s better to combine these two figures, where Figure6a is a better image to put in. You can mention that majority of the precipitates identified are shown in Figure 5.

-Fig.5b and 6b:  the Figure captions for (b) are wrong. They should be called selected area diffraction pattern.  In addition, these diffraction patterns are not proper.  They are not proper zone axis patterns, i.e. sample was not tiled properly to have a zone axis pattern.  Please pride the zone axis patterns.  The precipitates are big enough to tilt to zone axis. The precipitates were identified to have Al, N, Mn, Si. The authors identified the precipitates as aluminum nitrides with Si and Mn. I would not call AlN is an intermetallic compound.

The Cr is on the edge of the precipitates and is not inside it, but the authors described it to be inside.

The plane indexes are not clearly labeled and diffractions from the steel matix should be there, which were not mentioned.

Author Response

Question: «This translates into significant reductions in carburizing process time» (222-223). Can you specify the time?

  • when comparing the carburization processes presented in this article, the reduction in carburization time for the parameters assumed here was 127 minutes in favor of PreNitLPC®. The sentence was included in the article. (Line 293-295).

Reviewer 2

This manuscript reports the microstructure difference of low-pressure carburizing process with and without pre-nitriding. It is clearly written, and I suggest acceptance with revisions.

-In abstract, line 14: “on its basis”;  an incorrect word?  What do you intend to say? Base?

- thank you for pointing it out - the incorrectly used wording has been corrected in the abstract.

-Line 60: “intermetallic phases on its base”. Please clarify. Does the intermetallic phases form on the surface or inside core of the steel? What do you mean by base?

- thank you for noticing, the wording has been changed. Nitrides are formed in the surface layer of steel. In the temperature range 400-700, nitrogen dissolves into ferrite and forms nitrides with aluminum at higher temperatures.

-Line 93: Delete “Assumed”.

- we corrected

-Figure 4:  it will be better if you can add two more images into this figure: the SEM grain images of the surface of the steel of samples with and without pre-nitriding.

- we have added S/TEM images (Figure 6) of the steel surface layer and the microstructure of the steel after etching in Mi1Fe reagent (Figure 4). SEM images of the steel surface are not available to us, and we have no way of completing the study in this regard. Nevertheless, the optical microscope images of the cross-section sufficiently show the effect of grain grinding.

-Can the author provide SEM images to show where are the intermetallic precipitates located?  This image could be the same image in Figure 4.

- we have included STEM images of the precipitated intermetallic phases in Figure 6 and inserted measurements of their content in the steel after LPC and after PreNitLPC - Table 3.

- Figure 5 and Figure 6, and Line 196-197: “Therefore, it was decided to include in the article results two examples of the identified precipitates to show that they are of the same type.”  Please rephrase this sentence. It’s better to combine these two figures, where Figure6a is a better image to put in. You can mention that majority of the precipitates identified are shown in Figure 5.

– we have made changes to the article as suggested.

-Fig.5b and 6b:  the Figure captions for (b) are wrong. They should be called selected area diffraction pattern. 

- we corrected as suggested.

In addition, these diffraction patterns are not proper.  They are not proper zone axis patterns, i.e. sample was not tiled properly to have a zone axis pattern.  Please pride the zone axis patterns.  The precipitates are big enough to tilt to zone axis.

- The quality of these diffraction patterns may not be the best, but this does not change the diffraction image so much that the precipitates cannot be identified from it. The purpose of this study was to identify the precipitates formed that block the growth of austenite grains. Based on the obtained and resolved diffraction images of the precipitates, it can be concluded that the goal was achieved.

The precipitates were identified to have Al, N, Mn, Si. The authors identified the precipitates as aluminum nitrides with Si and Mn. I would not call AlN is an intermetallic compound.

-Schulze in 1967 defined intermetallic compounds as solid phases containing two or more metallic elements, with optionally one or more non-metallic elements, whose crystal structure differs from that of the other constituents. (G. E. R. Schulze: Metallphysik, Akademie-Verlag, Berlin 1967)

The Cr is on the edge of the precipitates and is not inside it, but the authors described it to be inside.

- That's right, we made a mistake, Cr is not inside the divisions. This has been corrected in the article.

The plane indexes are not clearly labeled and diffractions from the steel matix should be there, which were not mentioned.

- The nanobeam diffraction (NBD) method was used for testing, which is why, in our opinion, no reflexes from the steel matrix are visible

Reviewer 3 Report

Comments and Suggestions for Authors

The article "Identification of intermetallic phases limiting the growth of austenite grains in the low-pressure carburizing process" is interesting, the studies are sufficiently explained, the figures are relevant, therefore I consider that the work can be published

Author Response

The article "Identification of intermetallic phases limiting the growth of austenite grains in the low-pressure carburizing process" is interesting, the studies are sufficiently explained, the figures are relevant, therefore I consider that the work can be published.

  • Thank you for your very positive review.

Round 2

Reviewer 2 Report

Comments and Suggestions for Authors

Reviewer comments: In addition, these diffraction patterns are not proper.  They are not proper zone axis patterns, i.e. sample was not tiled properly to have a zone axis pattern.  Please pride the zone axis patterns.  The precipitates are big enough to tilt to zone axis.

Author reply- The quality of these diffraction patterns may not be the best, but this does not change the diffraction image so much that the precipitates cannot be identified from it. The purpose of this study was to identify the precipitates formed that block the growth of austenite grains. Based on the obtained and resolved diffraction images of the precipitates, it can be concluded that the goal was achieved.

Second round reviewer comments:

I agree that the precipitates contains Al, N, Si Mn from EDS map, but I don’t think you identified the precipitates correctly at all.

The diffraction pattern you indexed in Fig. 7 and Fig. 8 are not correct.

First, in Fig.7b, the stronger diffraction spots should be from the steel matrix, and you can see the weaker diffraction spots which are probably from the precipitates. But you can not index this pattern as AlN [-1010] because the angle between [000-1] and (0-110) should be 90 degree, not the 94.7 degree from your experimental diffraction pattern.

Second, in Fig 8b, the strong bright spots are from matrix, and the very weak ones looking like lines are probably from the precipitate. In addition, the planes you labeled are not from [0-110] zone axis. The planes from this zone axis should be (0002), (2-1-10), or (2-1-22) etc.

So it is not right that you identified AlN from these diffraction patterns.

I would suggest:

-You acquire proper diffraction pattern from the precipitates and index them correctly.

-Or you delete these diffraction data from the manuscript and only talks about these precipitates based on their compositions.

Author Response

Dear Mr. Reviewer

Thank you for your valuable comments and suggestions. As suggested, we have removed the part about diffraction from the article. We hope that everything is OK now.

Best regards

Leszek Klimek, Konrad Dybowski
